# Some Model Theoretic Properties for Pavelka-Style Gödel Logic, RGL* and Gödel Logic with Δ

**Nazanin Roshandel Tavana**

Department of Mathematics and Computer Science, Amirkabir University of Technology, Tehran 11366, Iran; nrtavana@aut.ac.ir

**Abstract:** Pavelka-style (rational) Gödel logic is an extension of Gödel logic which is denoted by RGL*. In this article, due to the approximate Craig interpolation property for RGL*, the Robinson theorem and approximate Beth theorem are presented and proved. Then, the omitting types theorem for this logic is expressed and proved. At the end, as a reduction, the omitting types theorem for standard Gödel logic with Δ is studied.

**Keywords:** Robinson consistency theorem; Beth theorem; omitting types theorem; rational Gödel logic; Gödel logic with Δ

## 1. Introduction

The following article contains original research on fuzzy set theory in a narrow sense. Although Gödel logic is originally a non-classical logic, it can be extended to other non-classical logics by adding appropriate nullary or unary connectives. This feature makes it possible to have new properties which help to prove mathematical theorems in these logics. It is worth noting that these theorems may not be proved in Gödel logic due to the absence of these features. For instance, in RGL*, by adding rational numbers as nullary connectives, one can determine the area of a formula before a value is assigned to it. This helps to prove a version of some theorems, such as the Craig interpolation property and its applications. Moreover, by adding a unary connective, Δ, to Gödel logic, one can determine the value of a (closed) formula as being either absolutely true or not, which also helps to prove some properties for formulas and theories. The theorems studied in this article are interesting for mathematicians in logic (model theory), and in particular in the model theory of fuzzy logics.

The Craig interpolation property is an open problem for first-order Gödel logic [1]. In [2], the author expresses an approximate version of the Craig interpolation property for Pavelka-style Gödel logic, RGL* (which is called rational Gödel logic). It says that, for every two sentences $\varphi$ and $\psi$ in the languages $\mathcal{L}_1$ and $\mathcal{L}_2$ such that $\varphi \models \psi$, and every $s \in (0,1] \cap \mathbb{Q}$, there is a sentence $\theta$ in $\mathcal{L}_1 \cap \mathcal{L}_2$ such that $\varphi \models \theta$ and $\theta \models \bar{s} \to \psi$. Note that, in this extension of Gödel logic, the rational numbers in $[0,1]$ are added to the language as nullary connectives as $\bar{s}$ for every $s \in [0,1] \cap \mathbb{Q}$. Also, the set of truth values in this new logic is $\mathbb{I} = [0,1]^2 \setminus \{(0,r) : r > 0\}$ for compactness. The value of every $\bar{s}$ for $s \in [0,1] \cap \mathbb{Q}$ in every standard model is given by $(s,s) \in \mathbb{I}$. The relation of RGL* to Gödel logic is the same as Pavelka logic to Łukasiewicz logic. The difference is that every operation as well as the implication in (Pavelka) Łukasiewicz logic is continuous, while the implication in Gödel logic (as well as RGL*) is not continuous. The model theory of this logic is similar to the metric model theory [3,4], by considering the concept of approximation in defining the notions.

As some applications of this version of the Craig interpolation property, the Robinson consistency theorem and an approximate version of the Beth theorem can be proved in RGL*.

The Craig interpolation property and its direct application as the Robinson consistency theorem are applicable in computing sciences in particular, databases [5], automated reasoning [6], type checking [7], model checking [8], structured theorem proving [9], etc. Computing science and model theoretic motivations have led to a very general approach to interpolation [10], which is completely independent of any concrete logical system [11–13].

The Robinson consistency theorem for Łukasiewicz logic in [14,15] as well as for many-valued logic in [16] is studied, but efforts for proving this theorem in Gödel logic have not been successful yet. In the presented article, the mentioned theorem in RGL* is formulated and proved. The Robinson consistency theorem in RGL* is similar to its classical version. It says that if $\mathcal{L}_1$ and $\mathcal{L}_2$ are two languages and $T$ is a linearly complete theory in $\mathcal{L}_1 \cap \mathcal{L}_2$, then for every two strongly consistent theories $T_1$ and $T_2$ in $\mathcal{L}_1$ and $\mathcal{L}_2$, respectively, both of them include $T$, the theory $T_1 \cup T_2$ is also strongly consistent in $\mathcal{L}_1 \cup \mathcal{L}_2$.

Moreover, an approximate version of the Beth theorem due to the approximate nature of this logic, similar to the approximate Craig interpolation property, can be proved. Assume $P$ and $P'$ are two new predicate symbols not in the language $\mathcal{L}$, $\Sigma(P)$ is the set of sentences in $\mathcal{L} \cup \{P\}$, and $\Sigma(P')$ is the set of sentences in $\mathcal{L} \cup \{P'\}$ by replacing $P$ by $P'$ everywhere. Then, if $\Sigma(P)$ *proves $P$ implicitly*, i.e.,

$$\Sigma(P) \cup \Sigma(P') \vdash \forall x_1, \ldots, x_n \ (P(x_1, \ldots, x_n) \leftrightarrow P'(x_1, \ldots, x_n))$$

then $\Sigma(P)$ *approximately defines $P$ explicitly*, i.e., for all $s \in (0,1]_{\mathbb{Q}}$, there is a sentence $\theta(x_1, \ldots, x_n)$ such that

$$\Sigma(P) \cup \{\forall x_1, \ldots, x_n \ \theta(x_1, \ldots, x_n)\} \models \forall x_1, \ldots, x_n \ P(x_1, \ldots, x_n)$$

and

$$\Sigma(P) \cup \{\forall x_1, \ldots, x_n \ P(x_1, \ldots, x_n)\} \models \bar{s} \rightarrow \forall x_1, \ldots, x_n \ \theta(x_1, \ldots, x_n).$$

It is proved that in RGL*, the approximate Beth theorem holds.

The model theory of fuzzy logic is introduced in [17–20]. The concept of type is crucial in the classical model theory, which is a collection of formulas which is satisfiable with respect to a theory [21]. Using a formula, one can provide a first-order property, and using a type, a collection of properties is presented. A type in a model $\mathcal{M}$ is *realized* if there are some elements $\bar{a}$ in the universe of $\mathcal{M}$ such that every formula substituting variables by $\bar{a}$ satisfies in $\mathcal{M}$. Otherwise, it is *omitted* in $\mathcal{M}$. The omitting types theorem says that, under some conditions, a type can be omitted in a model.

The omitting types theorem for some fuzzy logics is studied in [22–25]. In [22,23], the omitting types theorem is studied for $[0,1]$-valued logics in uncountable language, while in this article, the mentioned theorem is verified for $\mathbb{I}$-valued Gödel logic in countable language. Also, in Section 4, this theorem is mentioned and proved in a countable language. In [25], this theorem is proved for MTL, core fuzzy logics, and uniform logic, which are different from the logics presented here. Moreover, the approach used in the article [25] is via the proof theory, whereas the approach in this article is model theory. In comparison with [24], the usual completeness theorem does not hold in RGL*; furthermore, strong consistency is used instead of consistency. It is worth noting that strong consistency implies consistency but not vice versa. A theory $T$ is strongly consistent if for every $r > 0$, $T \nvdash \bar{r}$, while $T$ is called consistent if $T \nvdash \bar{1}$. In this new extension, due to the similarity of RGL* to the metric model theory, $\bar{0}$ is an absolute truth value, whereas $\bar{1}$ is an absolute false value. In Section 3.2, the omitting types theorem in RGL* is formulated and proved.

At the end, a proof for this theorem in Gödel logic with $\Delta$ is presented. A proof of the omitting types theorem in the logics with $\Delta$ is mentioned as a question at the end of [25]. The unary connective $\Delta$ gives the following property: For every closed formula $\varphi$, the value of $\Delta\varphi$ in every model in Gödel logic is 1 if the value of $\varphi$ is 1, and 0 otherwise.

In the preliminaries, the basic definitions and notions of rational Gödel logic are briefly introduced. In Section 3.1, first, the Robinson consistency theorem and approximate Beth theorem for RGL* are proved. Second, in the Section 3.2, the omitting types theorem for

this logic is presented. In the last section, Section 4, the standard Gödel logic with the values in $[0, 1]$ by an additional connective $\Delta$ is considered. As mentioned at the end of [25], it is worth verifying the omitting types theorem for logics with $\Delta$. Note that, despite the proof theoretic approach in [25], the approach which is used in this article is model theory. Thank to the properties which $\Delta$ provides for Gödel logic, the omitting types theorem can be proved for first-order Gödel logic with $\Delta$ by Henkin construction.

## 2. Preliminaries

Let $\mathcal{L}$ be a countable first-order language consisting of predicate, function and constant symbols, a countable set of variables, the quantifiers $\{\forall, \exists\}$, and a set of Boolean connectives $\{\wedge, \vee, \rightarrow, \neg\}$. Moreover, this language includes a set of nullary connectives $\{\bar{r} : r \in [0,1]_{\mathbb{Q}}\}$, where $[0,1]_{\mathbb{Q}} = [0,1] \cap \mathbb{Q}$. This is the reason for which this logic is called rational Gödel logic.

Despite the standard semantics of Gödel logic, the set of truth values is $\mathbb{I} = [0,1]^2 \backslash \{(0,r) : r > 0\}$ with lexicographical ordering to have the compactness theorem in rational Gödel logic, RGL* [26].

The concepts of an $\mathcal{L}$-structure, an assignment, $\mathcal{L}$-terms, interpretations of terms, and $\mathcal{L}$-formulas are defined as usual; see Definitions 2.1 and 2.2 in [26]. The interpretation of $\mathcal{L}$-formulas is defined as follows:

**Definition 1** (Definition 2.3 in [26]). *For an $\mathcal{L}$-structure $\mathcal{M}$ and an $\mathcal{M}$-assignment $\sigma$,*

1. *For every $\bar{r} \in [0,1]_{\mathbb{Q}}$, $\bar{r}^{\mathcal{M},\sigma} = (r,r) = \hat{r}$. Particularly, $\bar{1}^{\mathcal{M},\sigma} = \hat{1}$ and $\bar{0}^{\mathcal{M},\sigma} = \hat{0}$,*
2. $P^{\mathcal{M},\sigma}(t_1(\bar{x}), \ldots, t_n(\bar{x})) = P^{\mathcal{M}}(t_1^{\mathcal{M},\sigma}(\bar{x}), \ldots, t_n^{\mathcal{M},\sigma}(\bar{x})),$
3. $(\varphi \wedge \psi)^{\mathcal{M},\sigma}(\bar{x}) = \max\{\varphi^{\mathcal{M},\sigma}(\bar{x}), \psi^{\mathcal{M},\sigma}(\bar{x})\},$
4. $(\varphi \rightarrow \psi)^{\mathcal{M},\sigma}(\bar{x}) = \begin{cases} 0 & \varphi^{\mathcal{M},\sigma}(\bar{x}) \geq \psi^{\mathcal{M},\sigma}(\bar{x}), \\ \psi^{\mathcal{M},\sigma}(\bar{x}) & \varphi^{\mathcal{M},\sigma}(\bar{x}) < \psi^{\mathcal{M},\sigma}(\bar{x}). \end{cases}$
5. $(\forall x\, \varphi(x))^{\mathcal{M},\sigma} = \sup\{\varphi^{\mathcal{M},\sigma'}(x) : \sigma(x) = \sigma'(x)\},$
6. $(\exists x\, \varphi(x))^{\mathcal{M},\sigma} = \inf\{\varphi^{\mathcal{M},\sigma'}(x) : \sigma(x) = \sigma'(x)\},$

The connectives $\vee, \neg, \leftrightarrow$ are defined by the above connectives.

**Definition 2.** *For an $\mathcal{L}$-structure $\mathcal{M}$ and an $\mathcal{L}$-formula $\varphi(\bar{x})$ and an $\mathcal{L}$-theory $T$,*

1. *If $\varphi^{\mathcal{M}}(\bar{a}) = \hat{0}$ then $\varphi(\bar{x})$ is satisfied by $\bar{a} \in M$. It is denoted by $\mathcal{M} \models \varphi(\bar{a})$.*
2. *If $\mathcal{M} \models \psi$ for every $\psi \in T$, then the theory $T$ is satisfiable in $\mathcal{M}$, and indicated by $\mathcal{M} \models T$.*
3. *We say that $T$ entails $\varphi$ and denoted by $T \models \varphi$ if every $\mathcal{L}$-structure $\mathcal{M} \models T$ implies $\mathcal{M} \models \varphi$.*

As usual, satisfiability can be defined finitely.

**Remark 1.** *In the above definition, $\hat{0}$ is the absolute truth value, since the semantics of this logic is similar to the metric model theory.*

**Theorem 1.** *Let $u : [0,1]^2 \rightarrow [0,1]^2$ defined by $u(x,y) = (1-x, 1-y)$ and $\mathbb{I}^* = [0,1]^2 \backslash \{(1,r) : r < 1\}$. The function $u$ helps us to define an $\mathbb{I}^*$-interpretation in the language $\mathcal{L}$. By using this function, all semantical issues of RLG*, e.g., satisfiability and entailment, can be translated into fuzzy first-order rational Gödel logic. Hence, all the results are valid for fuzzy first-order rational Gödel logic.*

The proof system for RGL* was expressed in Section 2.1 in [26]. This proof system has some axioms for propositional Gödel logic, quantifiers, and some book-keeping axioms:

- (RGL1) $\bar{r} \wedge \bar{s} \leftrightarrow \overline{\max\{r,s\}}$,
- (RGL2) $\begin{cases} \bar{r} \rightarrow \bar{s} & \text{if } r \geq s, \\ (\bar{r} \rightarrow \bar{s}) \leftrightarrow \bar{s} & \text{if } r < s, \end{cases}$
- (RGL3) $\neg\neg\bar{r}$, for all $r < 1$.

*Modus ponens* and *generalization* are the inference rules. One can define the notion of proof as usual. Whenever an $\mathcal{L}$-theory $T$ *proves* an $\mathcal{L}$-sentence $\varphi$, it is denoted by $T \vdash \varphi$. The theory $T$ is called *consistent* if $T \nvdash \bar{1}$. Otherwise, it is called *inconsistent*. The deduction theorem is Theorem 2.5 in [26].

**Definition 3.** *An $\mathcal{L}$-theory $T$ is called strongly consistent if $T \nvdash \bar{r}$ for every $r > 0$.*

It is clear that for every $0 < r < 1$, $T = \{\bar{r}\}$ is consistent but it is not strongly consistent.

The soundness theorem can be proved in RGL*, i.e., if $T \vdash \varphi$, then $T \models \varphi$, for every $\mathcal{L}$-theory $T$ and $\mathcal{L}$-sentence $\varphi$. Thus, according to Remark 2.10 in [26], if a theory $T$ is satisfiable in an $\mathcal{L}$-structure $\mathcal{M}$, then $T$ is strongly consistent.

Now, the definition of a maximally strongly consistent theory and one of its characterizations are presented [26].

**Definition 4.** *A strongly consistent theory $T$ is called* maximally strongly consistent *if it is not a proper subset of any strongly consistent theory, i.e., for any strongly consistent theory $\Sigma$, if $T \subseteq \Sigma$, then $T = \Sigma$.*

**Lemma 1.** *Assume $T$ is a strongly consistent theory. Then, $T$ is maximally strongly consistent if and only if*

1.  *For every two sentences $\varphi$ and $\psi$, one of $\varphi \to \psi \in T$ or $\psi \to \varphi \in T$ holds.*
2.  *Assume $\varphi$ is a sentence. If for every $r \in (0,1]_{\mathbb{Q}}$, $T \vdash \bar{r} \to \varphi$, then $\varphi \in T$.*

**Proposition 1.** *If $T$ is a maximally strongly consistent theory such that for every $r \in (0,1]_{\mathbb{Q}}$, $T \models \bar{r} \to \varphi$ for a sentence $\varphi$, then $T \models \varphi$.*

**Proof.** Let, for a maximally strongly consistent theory $T$, a sentence $\varphi$ and every $r \in (0,1]_{\mathbb{Q}}$, $T \models \bar{r} \to \varphi$ and $T \nvDash \varphi$. Then, $T \cup \{\varphi\}$ is strongly consistent. Otherwise, assume $T \cup \{\varphi\} \vdash \bar{s}$ for some $s \in (0,1]_{\mathbb{Q}}$. Therefore, by the deduction theorem, $T \vdash \varphi \to \bar{s}$. By the soundness theorem, $T \models \varphi \to \bar{s}$. Therefore, $T \models \bar{r} \to \bar{s}$, for every $r \in (0,1]_{\mathbb{Q}}$. Thus, for every model $\mathcal{M} \models T$, it is deduced that $\hat{r} \geq \hat{s}$ for every $r \in (0,1]_{\mathbb{Q}}$, which is a contradiction.

Therefore, $T \cup \{\varphi\}$ is a strongly consistent theory including $T$. It contradicts the maximally strong consistency of $T$.  $\square$

**Lemma 2.** *For every strongly consistent theory $T$, there is a maximally strongly consistent theory which includes $T$.*

Below, a linear-complete and a Henkin-complete theory are defined. At the end of this section, a weak version of the completeness theorem is introduced, as seen in [26].

**Lemma 3.** *Assume $T$ is a strongly consistent $\mathcal{L}$-theory in RGL* and $\varphi, \psi \in Sent(\mathcal{L})$. Then, one of $T \cup \{\varphi \to \psi\}$ or $T \cup \{\psi \to \varphi\}$ is strongly consistent.*

**Definition 5.** *A theory $T$ is called* linear-complete *if for any two sentences $\varphi, \psi$, either $T \vdash \varphi \to \psi$ or $T \vdash \psi \to \varphi$.*

**Definition 6.** *An $\mathcal{L}$-theory $T$ is called Henkin-complete whenever $T \nvdash \forall x\, \varphi(x)$ there exists a constant symbol $c \in \mathcal{L}$ such that $T \nvdash \varphi(c)$.*

**Theorem 2** (Completeness). *In RGL*, every strongly consistent theory is satisfiable.*

**Corollary 1** (Compactness). *An $\mathcal{L}$-theory is finitely satisfiable if and only if it is satisfiable.*

This section ends with the approximate Craig interpolation property; see [2].

**Theorem 3** (Approximate Craig interpolation property)**.** *Let $\varphi$ and $\psi$ be two sentences in the languages $\mathcal{L}_1$ and $\mathcal{L}_2$, respectively. If $\varphi \models \psi$ then for every $s \in (0,1]_{\mathbb{Q}}$ there is a sentence $\theta$ in $\mathcal{L}_1 \cap \mathcal{L}_2$ such that $\varphi \models \theta$ and $\theta \models \bar{s} \to \psi$.*

## 3. Some Model Theoretic Properties for RGL*

　　This section is divided into two parts. In the first part, the Robinson consistency theorem and approximate Beth theorem as the applications of approximate Craig interpolation property [2], are presented. Then, in the next part, the omitting types theorem for RGL* is studied.

### 3.1. Robinson Consistency and Beth Theorems

　　As an application of approximate Craig interpolation property for rational Gödel logic in [2], first, the Robinson consistency theorem is presented. Despite the approximated version of Craig interpolation, this theorem can be proved as its classical version for RGL*.

**Theorem 4** (Robinson consistency theorem)**.** *Let $T$ be a linear complete theory in $\mathcal{L} = \mathcal{L}_1 \cap \mathcal{L}_2$, for two languages $\mathcal{L}_1$ and $\mathcal{L}_2$. If $T_1 \supseteq T$ and $T_2 \supseteq T$ are two strongly consistent theories in $\mathcal{L}_1$ and $\mathcal{L}_2$, respectively, then, $T_1 \cup T_2$ is a strongly consistent theory in $\mathcal{L}_1 \cup \mathcal{L}_2$.*

**Proof.** Let $T_1 \cup T_2$ be not strongly consistent. Therefore, there is $r \in (0,1]_{\mathbb{Q}}$ such that $T_1 \cup T_2 \vdash \bar{r}$. By Lemma 2, there are maximally strongly consistent theories $\hat{T}_1 \supseteq T_1$ and $\hat{T}_2 \supseteq T_2$. Therefore, $\hat{T}_1 \cup \hat{T}_2 \vdash \bar{r}$. Then, there are finite subsets $\Delta_1 \subseteq \hat{T}_1$ and $\Delta_2 \subseteq \hat{T}_2$ such that $\Delta_1 \cup \Delta_2 \vdash \bar{r}$. Assume $\sigma_1$ is the conjunction of the sentences in $\Delta_1$ and $\sigma_2$ is the conjunction of the sentences in $\Delta_2$. Thus, $\sigma_1 \wedge \sigma_2 \vdash \bar{r}$. Therefore, by Theorem 2.5 in [26], $\sigma_1 \vdash \sigma_2 \to \bar{r}$. The soundness theorem implies $\sigma_1 \models \sigma_2 \to \bar{r}$. Let $s \in (0,1]_{\mathbb{Q}}$ be such that $s < r$. By approximate Craig interpolation property for RGL*, there is $\theta$ in $\mathcal{L}$ such that $\sigma_1 \models \theta$ and $\theta \models \bar{s} \to (\sigma_2 \to \bar{r})$. Since $s < r$, the latter one implies $\theta \models \sigma_2 \to \bar{r}$. Hence, $\hat{T}_1 \models \theta$ which yields for all $t \in (0,1]_{\mathbb{Q}}$, $\hat{T}_1 \nvDash \theta \to \bar{t}$. Moreover, for all $t \in (0,1]_{\mathbb{Q}}$, $T \nvDash \theta \to \bar{t}$. On the other hand, $\hat{T}_2 \nvDash \theta$. Otherwise, $\hat{T}_2 \models \sigma_2 \to \bar{r}$, but $\hat{T}_2 \models \sigma_2$. Thus, there exists $t_0 \in (0,1]_{\mathbb{Q}}$ such that $T \nvDash \bar{t}_0 \to \theta$. Otherwise, if for all $t \in (0,1]_{\mathbb{Q}}$, $T \models \bar{t} \to \theta$, then, $\hat{T}_2 \models \bar{t} \to \theta$. This implies, by Proposition 1, $\hat{T}_2 \models \theta$ which is a contradiction. Hence, $T \nvDash \bar{t}_0 \to \theta$ and $T \nvDash \theta \to \bar{t}_0$, which contradicts with the linear completeness of $T$. □

　　Now, we express the Beth theorem and prove it. Despite the Robinson theorem, here, there are some changes in the classical Beth theorem. Similar to the approximate Craig interpolation property, an approximate version of the Beth theorem can be presented and proved.

**Definition 7.** *Let $P$ and $P'$ be two new $n$–place relation symbols which are not in the language $\mathcal{L}$. Moreover, let $\Sigma(P)$ be the set of sentences in language $\mathcal{L} \cup \{P\}$ and $\Sigma(P)$ be the corresponding set of sentences in language $\mathcal{L} \cup \{P'\}$ such that, in every sentence, $P$ is replaced by $P'$. Then,*

1. *$\Sigma(P)$ proves $P$ implicitly if*

$$\Sigma(P) \cup \Sigma(P') \vdash \forall x_1, \ldots, x_n \ (P(x_1, \ldots, x_n) \leftrightarrow P'(x_1, \ldots, x_n)).$$

2. *$\Sigma(P)$ approximately defines $P$ explicitly if for all $s \in (0,1]_{\mathbb{Q}}$, there is a sentence $\theta(x_1, \ldots, x_n)$ such that*

$$\Sigma(P) \cup \{\forall x_1, \ldots, x_n \ \theta(x_1, \ldots, x_n)\} \models \forall x_1, \ldots, x_n \ P(x_1, \ldots, x_n)$$

　　*and*

$$\Sigma(P) \cup \{\forall x_1, \ldots, x_n \ P(x_1, \ldots, x_n)\} \models \bar{s} \to \forall x_1, \ldots, x_n \ \theta(x_1, \ldots, x_n).$$

**Theorem 5.** *In RGL*, if $\Sigma(P)$ proves $P$ implicitly, then $\Sigma(P)$ approximately defines $P$ explicitly.*

**Proof.** Let $\Sigma(P)$ prove $P$ implicitly. Assume $c_1, \ldots c_n$ are new constant symbols added to $\mathcal{L}$. Therefore,

$$\Sigma(P) \cup \Sigma(P') \vdash P(c_1, \ldots, x_n) \to P'(c_1, \ldots, c_n).$$

Then, there are finite subsets $\Delta_1 \subseteq \Sigma(P)$ and $\Delta_2 \subseteq \Sigma(P')$ such that

$$\Delta_1 \cup \Delta_2 \vdash P(c_1, \ldots, c_n) \to P'(c_1, \ldots, c_n).$$

Assume the conjunction of sentences in $\Delta_1$ and also in $\Delta_2$ are $\psi(P)$ and $\psi(P')$, respectively. Then,

$$\psi(P) \wedge \psi(P') \vdash P(c_1, \ldots, c_n) \to P'(c_1, \ldots, c_n).$$

By the deduction theorem, one can deduce

$$\psi(P) \wedge P(c_1, \ldots, c_n) \vdash \psi(P') \to P'(c_1, \ldots, c_n).$$

By the soundness theorem,

$$\psi(P) \wedge P(c_1, \ldots, c_n) \models \psi(P') \to P'(c_1, \ldots, c_n).$$

Let $s \in (0, 1]_{\mathbb{Q}}$. Then, by approximate Craig interpolation property, there is a sentence $\theta(c_1, \ldots, c_n)$ in the language $\mathcal{L} \cup \{c_1, \ldots, c_n\}$ such that

$$\psi(P) \wedge P(c_1, \ldots, c_n) \models \theta(c_1, \ldots, c_n)$$

and

$$\theta(c_1, \ldots, c_n) \models \bar{s} \to (\psi(P') \to P'(c_1, \ldots, c_n)).$$

Note that every model $(\mathcal{M}, R)$ in the language $\mathcal{L} \cup \{P', c_1, \ldots, c_n\}$ can be a model in the language $\mathcal{L} \cup \{P, c_1, \ldots, c_n\}$ by interpreting $P$ by $R$ instead of interpreting $P'$ by $R$. Therefore,

$$\theta(c_1, \ldots, c_n) \models \bar{s} \to (\psi(P) \to P(c_1, \ldots, c_n)).$$

Now, let $\mathcal{M} \models \theta(c_1, \ldots, c_n) \wedge \psi(P)$. Then, $(\theta(c_1, \ldots, c_n))^{\mathcal{M}} = 0$ and $(\psi(P))^{\mathcal{M}} = 0$. Then, $\hat{s} \geq (\psi(P) \to P(c_1, \ldots, c_n))^{\mathcal{M}}$ implies $(P(c_1, \ldots, c_n))^{\mathcal{M}} \geq \hat{0}$. In both cases, either $b(P(c_1, \ldots, c_n))^{\mathcal{M}} = \hat{0}$ or $(P(c_1, \ldots, c_n))^{\mathcal{M}} > \hat{0}$, one can deduce $(\bar{s} \to (P(c_1, \ldots, c_n))^{\mathcal{M}} = \hat{0}$. Therefore, $\theta(c_1, \ldots, c_n) \wedge \psi(P) \models \bar{s} \to (P(c_1, \ldots, c_n)$. Since $c_1, \ldots, c_n$ do not occur in $\psi(P)$,

$$\Sigma(P) \cup \{\forall x_1, \ldots, x_n\, \theta(x_1, \ldots, x_n)\} \models \forall x_1, \ldots, x_n\, P(x_1, \ldots, x_n)$$

and

$$\Sigma(P) \cup \{\forall x_1, \ldots, x_n\, P(x_1, \ldots, x_n)\} \models \bar{s} \to \forall x_1, \ldots, x_n\, \theta(x_1, \ldots, x_n).$$

$\square$

*3.2. The Omitting Types Theorem*

In this section, the omitting types theorem for RGL* is expressed and proved.

First of all, the notion of a type is defined and the situation in which it is isolated is presented. According to the definition, the consistency of a type with respect to a theory is associated with its satisfiability up to an approximation.

In the rest of this article, a first-order language for RGL* is fixed and it is not usually mentioned in the concepts.

**Definition 8.**

1. *Assume $T$ is a theory. A set of $\mathcal{L}$-formulas with $n$ free variables is called an $n$-type consistent with $T$ if for every $\varepsilon \in (0, 1]_{\mathbb{Q}}$,*

$$T \cup \{\varepsilon \to \psi(\bar{x}) : \psi(\bar{x}) \in p\}$$

*is satisfiable.*

2. *The set of all linear-complete n-types is denoted by $S_n(T)$.*
3. *An $\mathcal{L}$-formula $\varphi(\bar{x})$ isolates an n-type p if for all $\psi \in p$ and $r \in (0,1]_{\mathbb{Q}}$,*

$$T \cup \varphi(\bar{x}) \models \bar{r} \to \psi(\bar{x}).$$

4. *An n-type p is realized in a structure $\mathcal{M}$ if there is $\bar{a} \in \mathcal{M}$ such that for every $\psi(\bar{x}) \in p$ and $r \in (0,1]_{\mathbb{Q}}$,*

$$\mathcal{M} \models \bar{r} \to \psi(\bar{a}).$$

**Theorem 6.** *Note that since we add nonstandard values to the set of truth values, for every $r \in (0,1]_{\mathbb{Q}}, \hat{0} < x < \hat{r}$ does not imply that $x = \hat{0}$.*

The following proposition presents a condition in which an isolated type is realized.

**Proposition 2.** *If $\varphi(\bar{x})$ isolates p, then p is realized in any model of $T \cup \{\exists \bar{x} \; \varphi(\bar{x})\}$. Therefore, if $T \cup \{\exists \bar{x} \; \varphi(\bar{x})\}$ is satisfiable and T is linear-complete then every isolated type is realized.*

**Proof.** Let $\psi(\bar{x}) \in p$. Assume $\mathcal{M} \models T \cup \{\exists x \; \varphi(\bar{x})\}$. Therefore, there is $\bar{a} \in M$ such that $\varphi^{\mathcal{M}}(\bar{a}) = \hat{0}$. Since $\varphi(\bar{x})$ isolates p for every $\psi \in p$ and $r \in (0,1]_{\mathbb{Q}}$,

$$T \cup \varphi(\bar{x}) \models \bar{r} \to \psi(\bar{x})$$

and since $\mathcal{M} \models T \cup \varphi(\bar{a})$, one can conclude $(\bar{r} \to \psi(\bar{a}))^{\mathcal{M}} = \hat{0}$, for every $\psi \in p$ and $r \in (0,1]_{\mathbb{Q}}$. Therefore, p is realized in $\mathcal{M}$. $\square$

In the following theorem, the omitting types theorem in RGL* is proved.

**Theorem 7.** *Let $\mathcal{L}$ be a countable language and T be a strongly consistent $\mathcal{L}$-theory. If p is a non-isolated type over $\varnothing$, then there is a countable model $\mathcal{M}$ of T which omits p.*

**Proof.** Assume $\mathcal{C} = \{c_0, c_1, \dots\}$ is a set of new constants and $\mathcal{L}^* = \mathcal{L} \cup C$. A linear-complete $\mathcal{L}^*$-theory $T^*$ which contains T with witness property and a model $\mathcal{M} \models T^*$ will be built.

The construction will be arranged such that for all $d_1, \dots, d_n \in \mathcal{C}$, there are $\psi(\bar{x}) \in p$ and $s \in (0,1]_{\mathbb{Q}}$ such that $T^* \cup \{\psi(d_1, \dots, d_n) \to \bar{s}\}$ is satisfiable.

Let $\varphi_0, \varphi_1, \varphi_2, \dots$ be an enumeration of $\mathcal{L}^*$-formulas. A list of $\mathcal{L}^*$-sentences $\theta_0, \theta_1, \dots$ will be built such that for every $i \in \mathbb{N}$,

$$\theta_{i+1} \models \theta_i,$$

and $T \cup \{\theta_i\}$ is satisfiable.

In stage 0, let $\theta_0 = \forall x \; (\varphi_0(\bar{x}) \to \bar{0})$. Now, let $\theta_s$ be built such that $T \cup \{\theta_s\}$ is satisfiable.

Stage $s + 1 = 3i + 1$: (linear-completeness) If $T \cup \{\theta_s \to \varphi_i\}$ is satisfiable then let $\theta_{s+1} := \theta_s \to \varphi_i$ and otherwise, if $T \cup \{\varphi_i \to \theta_s\}$ is satisfiable, then let $\theta_{s+1} := \varphi_i \to \theta_s$. If T is a strongly consistent theory then, according to Lemma 3, one of the above theories is strongly consistent and, therefore, satisfiable by Theorem 2.

Stage $s + 1 = 3i + 2$: (witness property) Let $\varphi_i = \forall x \; \psi(x)$.

By Lemma 2, there exists a maximally strongly consistent theory $\hat{T}$ that includes $T \cup \{\theta_s\}$. The set

$$T_i = \hat{T} \cup \{(\bar{r} \to \varphi_i) \vee (\psi(c) \to \bar{s}) : s < r, r, s \in (0,1]_{\mathbb{Q}}\}$$

is strongly consistent, where c is not used in $\hat{T}$ and $\varphi_i$. The proof is similar to the proof of claim 1 of Theorem 2.17 of [26]. Therefore, there is a maximally strongly consistent theory $T'$ that includes $T_i$. By Theorem 2, $T'$ is satisfiable. If $T' \nvdash \varphi_i$, by Lemma 1, there is

$r \in (0,1]_{\mathbb{Q}}$ such that $T' \nvdash \bar{r} \to \varphi_i$. Therefore, there exists $s \in (0,1]_{\mathbb{Q}}$ such that $T' \vdash \psi(c) \to \bar{s}$. By the soundness theorem, $T' \models \psi(c) \to \bar{s}$. Assume $\theta_{s+1} := \theta_s \wedge (\psi(c) \to \bar{s})$. Therefore, $T \cup \{\theta_{s+1}\}$ is satisfiable and $\theta_{s+1} \models \theta_s$.

Stage $s + 1 = 3i + 3$: (Omitting type) Let $\bar{d}_i = (e_1, \dots, e_n)$ and $\varphi(\bar{x})$ be constructed from $\theta_s$ by replacing each $e_i$ by $x_i$. Also, replace every $c \in \mathcal{C} \setminus \{e_1, \dots, e_n\}$ by $y_c$ and put $\exists y_c$ in front of the formula. Thus, we can discard all constants of $\theta_s$ and $\mathcal{C}$. Since $p$ is not isolated, there are $\psi(\bar{x}) \in p$ and $r \in (0,1]_{\mathbb{Q}}$ such that

$$T \cup \varphi(\bar{x}) \nvDash \bar{r} \to \psi(\bar{x}).$$

Therefore, there is a structure $\mathcal{M}$ and $\bar{a} \in M$ such that $\mathcal{M} \models T \cup \varphi(\bar{a})$ but $(\bar{r} \to \psi(\bar{a})^{\mathcal{M}} > \hat{0}$. Thus, $\hat{r} < \psi^{\mathcal{M}}(\bar{a})$. Then, by letting $\theta_{s+1} = \theta_s \wedge (\psi(\bar{d}_i) \to \bar{r})$, $T \cup \{\theta_{s+1}\}$ is satisfiable and $\theta_{s+1} \models \theta_s$.

Set $T^* = T \cup \{\theta_0, \theta_1, \dots\}$. By stages $3i + 1$ and $3i + 2$, $T^*$ is linear-complete and has witness property, respectively. Also, since in every stage $s$, $T \cup \{\theta_s\}$ is satisfiable, $T^*$ is satisfiable.

Therefore, as established by Theorem 2.20 of [26], there is a canonical model $\mathcal{M}$ of $T^*$. This model of $T^*$ omits $p$. This is due to the fact that every element $\bar{a}$ of $\mathcal{M}$ is the interpretation of some constant symbols, i.e., there are constant symbols $\bar{d}_i$ such that $\bar{d}_i^{\mathcal{M}} = \bar{a}$. At stage $3i + 3$, the construction is ensured such that $\mathcal{M} \models \psi(\bar{a}) \to \bar{r}$, for some $\psi(\bar{x}) \in p$ and $r \in (0,1]_{\mathbb{Q}}$. Therefore, $\mathcal{M}$ omits $p$. $\square$

**Theorem 8.** *Since in the above induction-style proof it is assumed that, in each stage $s$, $T \cup \{\theta_s\}$ is satisfiable, there is no need to verify $\exists$-Henkin for this construction, as mentioned in [25].*

**Example 1.** *Let $\mathcal{L} = \{R\}$ where $R$ is a unary predicate symbol. Assume that $\{r_i\}_{i \in \mathbb{N}}$ and $\{s_i\}_{i \in \mathbb{N}}$ are, respectively, increasing and decreasing sequences in $(0,1]_{\mathbb{Q}}$ such that $\lim_{i \to \infty} r_i = \lim_{i \to \infty} s_i$. Define an $\mathcal{L}$-theory*

$$T = \{\bar{s}_i \to \forall x\, R(x)\}_{i \in \mathbb{N}} \cup \{\forall x\, R(x) \to \bar{r}_i\}_{i \in \mathbb{N}}$$

*and $\mathcal{L}$-type $p(x) = \{r_i \to R(x)\}_{i \in \mathbb{N}}$. The theory $T$ is strongly consistent and satisfiable. For every $r \in (0,1]_{\mathbb{Q}}$ and every $i \geq 1$, $T \cup \{r_i \to R(x)\}_{i \in \mathbb{N}}$ is finitely satisfiable and by compactness it is satisfiable. Thus, $p$ is a 1-type as well as non-isolated. By the omitting types theorem, for every model $\mathcal{M}$, there are $r \in (0,1]_{\mathbb{Q}}$, $i \geq 1$ and $a \in M$ such that $\mathcal{M} \nvDash \bar{r} \to (\bar{r}_i \to R(a))$.*

## 4. Omitting Types Theorem for First-Order Gödel Logic with $\Delta$

In this section, the goal is to prove omitting types theorem in standard Gödel logic with $\Delta$ where the set of truth values is $[0,1]$. Since we return to (classical) fuzzy logic, here, the convention in the absolute truth value is 1 in spite of the previous sections in which the absolute truth value is 0. Note that, in the previous section, the method used is the metric model theory [3], while in this section the method is the (many-valued) model theory for fuzzy logics [17]. Note that, by Remark 1, in the previous section, the truth value can also be 1, but due to the framework of the metric model theory, it is assumed to be 0 [3].

First of all, a new unary connective symbol $\Delta$ is added to a first-order language. Then, a structure which is called a valuation in [27] is defined.

**Definition 9.** *A structure $\mathcal{M}$ into $[0,1]$ consists of*

1. *A non-empty set $M$ as the universe of $\mathcal{M}$;*
2. *For every $n$-ary predicate symbol $P$, $P^{\mathcal{M}} : M^n \to [0,1]$;*
3. *For every $n$-ary function symbol $f$, $f^{\mathcal{M}} : M^n \to M$;*
4. *For every term $t$, $t^{\mathcal{M}} \in M$.*

   *The cardinal of $\mathcal{M}$ is the same as the cardinal of its universe $M$.*

Next, the value of a sentence (a formula without any free variables) is defined.

**Definition 10.** *For a given structure $\mathcal{M}$, the value of a sentence $\varphi$, $\varphi^{\mathcal{M}}$, is defined as follows:*

1. *For an atomic formula $\varphi = P(x_1, \ldots, x_n)$, $\varphi^{\mathcal{M}} = P^{\mathcal{M}}(a_1, \ldots, a_n)$, where $a_1, \ldots, a_n \in M$.*
2. *For every sentences $\varphi$ and $\psi$,*

    *(a)* $(\bot)^{\mathcal{M}} = 0$,

    *(b)*
    $$(\Delta\varphi)^{\mathcal{M}} = \begin{cases} 1, & \varphi^{\mathcal{M}} = 1 \\ 0, & \varphi^{\mathcal{M}} < 1 \end{cases}$$

    *(c)* $(\varphi \wedge \psi)^{\mathcal{M}} = \min(\varphi^{\mathcal{M}}, \psi^{\mathcal{M}})$,

    *(d)* $(\varphi \vee \psi)^{\mathcal{M}} = \max(\varphi^{\mathcal{M}}, \psi^{\mathcal{M}})$,

    *(e)*
    $$(\varphi \to \psi)^{\mathcal{M}} = \begin{cases} 1, & \varphi^{\mathcal{M}} \leq \psi^{\mathcal{M}} \\ \psi^{\mathcal{M}}, & \varphi^{\mathcal{M}} > \psi^{\mathcal{M}} \end{cases}$$

    *(f)* $(\forall x\, \varphi(x))^{\mathcal{M}} = \inf\{(\varphi(a))^{\mathcal{M}} : a \in M\}$,

    *(g)* $(\exists x\, \varphi(x))^{\mathcal{M}} = \sup\{(\varphi(a))^{\mathcal{M}} : a \in M\}$.

    *For a sentence $\varphi$ and a structure $\mathcal{M}$, if $\varphi^{\mathcal{M}} = 1$, then $\mathcal{M}$ is a* model *of $\varphi$ or $\varphi$ is satisfiable in $\mathcal{M}$. Also, for a set of sentences $\Gamma$, if for each sentence $\varphi \in \Gamma$, $\varphi^{\mathcal{M}} = 1$, then $\mathcal{M}$ is a* model *of $\Gamma$.*

**Notation 1.** *For simplicity, let $\sim \varphi$ denote $\neg\Delta\varphi$, for every formula $\varphi$.*

Therefore,
$$(\neg\varphi)^{\mathcal{M}} = \begin{cases} 1, & \varphi^{\mathcal{M}} = 0 \\ 0, & \varphi^{\mathcal{M}} > 0 \end{cases}$$

and
$$(\sim \varphi)^{\mathcal{M}} = \begin{cases} 0, & \varphi^{\mathcal{M}} = 1 \\ 1, & \varphi^{\mathcal{M}} < 1 \end{cases}$$

Below, the notion of 1-entailment is given.

**Definition 11** ((1-entailment) Definition 14 of [27])**.** *A set $\Gamma$ of sentences* 1-entails *a sentence $\varphi$ and denoted by $\Gamma \Vdash \varphi$ if, for every interpretation (valuation) $\mathcal{M}$, whenever the interpretations (values) of all formulas in $\Gamma$ are 1, then the interpretation (value) of $\varphi$ is also 1.*

**Remark 2.** *According to Lemma 16, Theorems 22 and 25 in [27], soundness and completeness hold in Gödel logic with $\Delta$.*

**Theorem 9** (Compactness)**.** *A set of formulas $\Gamma$ is satisfiable if and only if every one of its finite subsets is satisfiable.*

**Proof.** By Lemma 16 and Theorems 22 and 25 of [27], it is easily proved. □

Next, we present the definitions of an *n*-type, its realization and its omitting in a structure. In this logic, these notions are accurate and similar to ones in classical logic [21], by the properties which $\Delta$ provides.

**Definition 12.** *A set $p$ of $\mathcal{L}$ with free variables $x_1, \ldots, x_n$ is called an n-type (with respect to a theory $T$) if $T \cup p$ is satisfiable. An n-type $p$ is $\Delta$-complete if $\varphi \in p$ or $\sim \varphi \in p$, for all formulas $\varphi$ with free variables in $x_1, \ldots, x_n$. An n-type $p$ is realized in a structure $\mathcal{M}$, if there is $\bar{a} = (a_1, \ldots, a_n) \in M$ such that for every $\varphi \in p$, $(\varphi(\bar{a}))^{\mathcal{M}} = 1$. Otherwise, $p$ is omitted in $\mathcal{M}$.*

In the following, first, the notions of $\Delta$-completeness and linear completeness are defined. Second, in a lemma, the relation between $\Delta$-completeness and linearly completeness is mentioned and proved. This lemma is used to construct a model which omits a non-isolated *n*-type.

**Definition 13.** *Let* $\Gamma$ *be a set of sentences.*

1. $\Gamma$ *is called* $\Delta$-*complete if for every sentence* $\varphi$ *in the language,* $\varphi \in \Gamma$ *or* $\sim \varphi \in \Gamma$.
2. $\Gamma$ *is called linearly complete if for every sentences* $\varphi$ *and* $\psi$, $\Gamma \Vdash \varphi \to \psi$ *or* $\Gamma \Vdash \psi \to \varphi$.

**Lemma 4.** *Let a set* $\Gamma$ *of sentences be* $\Delta$-*complete. Then,* $\Gamma$ *is linearly complete.*

**Proof.** Assume there are two sentences $\varphi$ and $\psi$ such that $\Gamma \nVdash \varphi \to \psi$. Then, by $\Delta$-completeness, $\Gamma \Vdash \sim (\varphi \to \psi)$. Therefore, for every model $\mathcal{M}$ of $\Gamma$, $(\sim (\varphi \to \psi))^{\mathcal{M}} = 1$. Therefore, $(\varphi \to \psi)^{\mathcal{M}} < 1$ which yields that $\varphi^{\mathcal{M}} > \psi^{\mathcal{M}}$ and $\psi^{\mathcal{M}} < 1$. Hence, $(\psi \to \varphi)^{\mathcal{M}} = 1$. Thus, $\Gamma \Vdash \psi \to \varphi$. $\square$

Now, it is time to define an isolated type.

**Definition 14.** *Let* $T$ *be an* $\mathcal{L}$-*theory and* $\varphi(x_1, \ldots, x_n)$ *be an* $\mathcal{L}$-*formula with n free variables such that* $T \cup \{\varphi(\bar{x})\}$ *is satisfiable and* $p(\bar{x})$ *be an n-type with regard to T. A formula* $\psi(\bar{x})$ *with n free variables* isolates *p if for every* $\varphi \in p$,

$$T \cup \{\psi(\bar{x})\} \Vdash \varphi(\bar{x}).$$

Note that, for a $\Delta$-complete $n$-type $p$, if $\psi(\bar{x})$ isolates $p$, then

$$T \cup \{\psi(\bar{x})\} \Vdash \varphi(\bar{x}) \longleftrightarrow \varphi(\bar{x}) \in p$$

for every $\mathcal{L}$-formula $\varphi(\bar{x})$. Since if $\varphi(\bar{x}) \notin p$, by $\Delta$-completeness, $\sim \varphi(\bar{x}) \in p$. Then, $T \cup \{\psi(\bar{x})\} \Vdash \varphi(\bar{x})$ by hypothesis and $T \cup \{\psi(\bar{x})\} \Vdash \sim \varphi(\bar{x})$ by isolation. Therefore, since $T \cup \{\psi(\bar{x})\}$ is satisfiable, there is a model $\mathcal{M}$ of $T \cup \{\psi(\bar{x})\}$ and the argument implies $(\varphi(\bar{a}))^{\mathcal{M}} = 1$ and $(\varphi(\bar{a}))^{\mathcal{M}} < 1$, for every $\bar{a} \in M$. This is a contradiction.

Next, the concept of witness property is defined, which is used in Henkin construction to build a structure which omits a non-isolated $n$-type. Note that, since the used approach is the model theory, this property is defined from a model theoretic point of view.

**Definition 15.** *An* $\mathcal{L}$-*theory T has witness property whenever* $T \nVdash \forall x \, \varphi(x)$; *then, there is a constant symbol* $c \in \mathcal{L}$ *such that* $T \nVdash \varphi(c)$.

The following proposition says that, if an $n$-type is isolated by a formula, then it is realized in every structure wherein there is a realization for that formula. Therefore, in the omitting types theorem, one should assume the given $n$-type is non-isolated. The proof is similar to the one in [21].

**Proposition 3.** *If* $\psi(\bar{x})$ *isolates an n-type* $p(\bar{x})$ *then* $p(\bar{x})$ *is realized in any model of* $T \cup \{\exists \bar{x} \, \psi(\bar{x})\}$. *Particularly, if T is* $\Delta$-*complete then every isolated type is realized.*

Finally, all of the above definitions and theorems can be followed by the main result, the omitting types theorem for first-order Gödel logic with $\Delta$.

**Theorem 10.** *Assume* $\mathcal{L}$ *is a countable language, T is an* $\mathcal{L}$-*theory and p is a (possibly* $\Delta$-*incomplete) non-isolated n-type over* $\varnothing$. *Then, there is a countable model* $\mathcal{M}$ *of T which omits p.*

**Proof.** Let $C = \{c_0, c_1, \ldots\}$ be a new set of constant symbols and $\mathcal{L}^* = \mathcal{L} \cup C$. The goal is to construct a $\Delta$-complete $\mathcal{L}^*$-theory $T^* \supseteq T$ with witness property such that for all $d_1, \ldots, d_n \in C$ there is a formula $\varphi(\bar{x}) \in p$ for which $T^* \Vdash \sim \varphi(d_1, \ldots, d_n)$ holds. Therefore, there is a model $\mathcal{M}$ of $T^*$ such that $d_1^{\mathcal{M}}, \ldots, d_n^{\mathcal{M}}$ do not realize $p$, because every element of $M$ is the interpretation of a constant symbol in $C$. Therefore, $p$ is omitted in $\mathcal{M}$.

A sequence $\theta_0, \theta_1, \ldots$ of $\mathcal{L}^*$-sentences will be constructed such that $\Vdash \theta_t \to \theta_s$ for all $t > s$ and $T^* = T \cup \{\theta_i\}_{i=0}^{\infty}$ will be a satisfiable extension of $T$.

Let $\varphi_0, \varphi_1, \dots$ be a list of all $\mathcal{L}^*$-sentences.

Stage 0: $\theta_0 := \forall x \, (\varphi(x) \to\sim \bot)$.

Assume $\theta_s$ is constructed such that $T \cup \{\theta_s\}$ is satisfiable.

$s + 1 = 3i + 1$: ($\Delta$-completeness) If $T \cup \{\theta_s, \varphi_i\}$ is satisfiable, then let $\theta_{s+1} := \theta_s \wedge \varphi_i$. Otherwise, let $\theta_{s+1} := \theta_s \wedge \sim \varphi_i$. Note that just one of the $T \cup \{\theta_s \wedge \varphi_i\}$ and $T \cup \{\theta_s \wedge \sim \varphi_i\}$ is satisfiable. Since $T \cup \{\theta_s\}$ is satisfiable, there is a model $\mathcal{M}$ of $T \cup \{\theta_s\}$. If $\varphi_i^{\mathcal{M}} = 1$, then $T \cup \{\theta_s \wedge \varphi_i\}$ is satisfiable by $\mathcal{M}$. Otherwise, $\varphi_i^{\mathcal{M}} < 1$ which implies $(\sim \varphi_i)^{\mathcal{M}} = 1$ and $T \cup \{\theta_s \wedge \sim \varphi_i\}$ is satisfiable by $\mathcal{M}$.

$s + 1 = 3i + 2$: (Witness property) Let $\varphi_i = \forall x \, \psi(x)$ and $T \cup \{\theta_s\} \nVdash \forall x \, \psi(x)$. Set $\theta_{s+1} = \theta_s \wedge \sim \psi(c)$ for a new constant symbol $c \in C$ which does not occur in $T \cup \{\theta_s\}$. Since $T \cup \{\theta_s\}$ is satisfiable, there is a model $\mathcal{M}$ of $T \cup \{\theta_s\}$. Therefore, $\inf_{a \in M}(\psi(a))^{\mathcal{M}} < 1$. Thus, there is $a_0 \in M$ such that $(\psi(a_0))^{\mathcal{M}} < 1$ and then, $((\sim \psi)(a_0))^{\mathcal{M}} = 1$. Therefore, $T \cup \{\theta_{s+1}\}$ is satisfiable by $\mathcal{M}$ with the interpretation of $c$ by $a_0$.

$s + 1 = 3i + 3$: (Omitting $p$) Let $\bar{d} = (e_1, \dots, e_n)$ and by changing constants and variables in $\theta_s$ as the omitting types theorem in [21], $\theta_s$ turns into a formula $\psi(\bar{x})$ with $n$ variables. Non-isolating $p$ implies that there is $\varphi(\bar{x}) \in p$ such that $T \cup \{\psi(\bar{x})\} \nVdash \varphi(\bar{x})$. Therefore, there is a structure $\mathcal{M}$ and $\bar{a}_0 \in M$ so that $\mathcal{M}$ is a model of $T \cup \{\psi(\bar{a}_0)\}$ and $(\varphi(\bar{a}_0))^{\mathcal{M}} < 1$. Thus, $(\sim \varphi(\bar{a}_0))^{\mathcal{M}} = 1$. It yields that, by assuming $\theta_{s+1} := \theta_s \wedge \sim \varphi(\bar{d})$, the theory $T \cup \{\theta_{s+1}\}$ is also satisfiable by $\mathcal{M}$ and interpreting $\bar{d}$ by $\bar{a}_0$.

Now, assume $T^* = T \cup \{\theta_i : i \geq 0\}$. By the construction and Theorem 9, $T^*$ is satisfiable and $\Delta$-complete (and so, by Lemma 4, linearly complete) with witness property. Let $\mathcal{M}$ be a canonical model of $T^*$ as in the proof of strong completeness in [27], but in that model the constant symbols are assumed as the elements of universe instead of terms. Then, $\mathcal{M}$ omits $p$; since every element of $M$ is the interpretation of a constant symbol, for every $\bar{a} \in M$, there are $\bar{d} \in C$ such that $\bar{d}^{\mathcal{M}} = \bar{a}$. Stage $3i + 3$ guarantees that $(\sim \varphi(\bar{a}))^{\mathcal{M}} = 1$ for some $\varphi \in p$. Therefore, $(\varphi(\bar{a}))^{\mathcal{M}} < 1$ and this structure omits $p$. $\square$

## 5. Concluding Remarks and Further Works

Rational Gödel logic is a Pavelka-style extension of Gödel logic. The introduction of the model theory of this logic began in [26]. In this article, first, two model theoretic theorems, namely the Robinson theorem and an approximate version of the Beth theorem due to approximate Craig interpolation property, are proved in RGL*. Then, in the resumption, the omitting types theorem is proved in this logic. Then, the logic is reduced to standard Gödel logic in which $\Delta$ is added to as a unary connective. Finally, the omitting types theorem is verified for Gödel logic with $\Delta$.

As a further work, one can study other model theoretic properties for these presented logics. Moreover, extending the results in this article to other areas such as intuitionistic fuzzy set theory and to generalized algebraic structures may be interesting.

**Funding:** This research received no external funding.

**Data Availability Statement:** There is no data used in this article.

**Acknowledgments:** I am grateful to the anonymous referees for providing very useful comments that have helped to greatly improve this paper.

**Conflicts of Interest:** The author declares no conflict of interest.

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
