# Peer review of "Some Model Theoretic Properties for Pavelka-Style Gödel Logic, RGL* and Gödel Logic with Δ"

_axioms, doi:10.3390/axioms12090858_

Round 1

Reviewer 1 Report

The paper is not well written. Need to improve before publication in the following way:

(1) Introduction need to revise, need to add more related references and compare the present work with respect to related work.

(2) Motivation and novelties should be mention in a separate section in introduction section.

(3) There is no equation numbers. Need to give equation numbers.

(4) Application is not clear. Need to give a brief not that what is the application of that work.

(5) There is only mathematical findings, very much difficult to find the main contribution. 

(6) Numerical study is missing.

(7) Conclusion and future research scope should be extended. 

(8) Comparative result should be illustrated. 

(9) Is the result is valid multivalued logic. 

Extensive editing of English language required

Reviewer 2 Report

The paper is interesting and in general correct, but I recommend to the authors to define briefly for readers' convenience  the symbols below:

- overline{s} in the Introduction, line 6 and below, 

- Delta - on page 2 and below.

In order to stick to the already introdiced notation for sets, it will be better if the expression in Example 3.9 that has the form

\{r_i\}^{\infty}_{i=1}

and the next ones, to have the form, e.g.,

\{r_i\ : i \in N \}.

 I recommend to the author in future to extend his/her results to the case of intuitionistic fuzzy sets, for which a variant of Craig's interpolation theorem exists.

Reviewer 3 Report

I recommend the author to supplement the motivation and other examples
of the studied structure.
Furthermore, the author could develop a discussion in the conclusion,
whether this kind of square can also be interpreted in generalized
algebraic structures.

Round 2

Reviewer 1 Report

The author revise the paper sucessfully. But need to format as per journal templete. 

Language editing needed.